

# Psychometric properties of the Multi-group Ethnic Identity Measure (MEIM) in a sample of Iranian young adults

Mojtaba Habibi[1,2,3], Maede Sadat Etesami[1,4,5], Mohammad Ali Taghizadeh[6], Faezeh Sadat Akrami[7] and Danilo Garcia[8,9,10,11,12]

[1] Health promotion Research Center, Iran University of Medical Sciences, Tehran, Iran
[2] Department of Health Psychology, School of Behavioral Sciences and Mental Health (Tehran Institute of Psychiatry), Iran University of Medical Sciences, Tehran, Iran
[3] Department of Psychology, Norwegian University of Science and Technology, Trondheim, Norway
[4] Department of Clinical Psychology, University of Tehran, Tehran, Iran
[5] Iranian Research Center for HIV/AIDS (IRCHA), Department of Infectious Diseases, Imam Khomeini Hospital Complex, Tehran, Iran
[6] Department of Counseling, Shaheed Beheshti University, Tehran, Iran
[7] Department of Clinical Psychology, School of Humanities and Social sciences, Islamic Azad University, Science and Research Branch, Tehran, Iran
[8] Department of Behavioral Sciences and Learning, Linköping University, Linköping, Sweden
[9] Center for Ethics, Law and Mental Health (CELAM), University of Gothenburg, Gothenburg, Sweden
[10] Promotion of Health and Innovation (PHI) Lab, Network for Well-Being, Sweden
[11] Department of Psychology, University of Gothenburg, Gothenburg, Sweden
[12] Blekinge Centre of Competence, Region Blekinge, Karlskrona, Sweden

## ABSTRACT

**Background.** This study examines the factor structure, reliability and test-retest validity of the 12-item Iranian version of the Multigroup Ethnic Identity Measure (MEIM). Additionally, the MEIM's concurrent validity was tested by investigating the association between ethnic identity and subjective well-being.

**Method.** The scale was translated into Persian language and was administered to 426 students (193 female) at a major public university in Tehran along with the Positive Affect Negative Affect Schedule, and the Satisfaction with Life Scale.

**Results.** The confirmatory factor analysis supported the two-factor first-order commitment, and exploration (consisting of 12 items), and the second-order unidimensional factor structure of general ethnic identity. Moreover, we found evidence for good internal consistency, test re-test reliability, and concurrent validity.

**Conclusion.** The MEIM Persian version was found to be a valid and reliable measure to examine ethnic identity in this Iranian student population, for both males and females. These results support the utility of the Persian version of the MEIM for its use in Middle-Eastern contexts.

## INTRODUCTION

Ethnic identity is a multidimensional and ever-evolving phenomena that has been defined in various ways. *Tajfel (1981)*, for example, defined ethnic identity as part of one's individuality

Corresponding authors
Maede Sadat Etesami,
etesami.m@ums.ac.ir
Danilo Garcia,
danilo.garcia@icloud.com

that comes from the understanding of one's membership in an ethnic group with particular values and emotions. Similarly, *Helms (2007)* defined it as belonging to an ethnic group that has specific heritage, values and characteristics (*Phinney & Ong, 2007*). Regardless of the definition, ethnic identity is a component of the question: "who am I?", and therefore a powerful factor in the development and preservation of ethnic groups and their social bonds. At the same time, its formation is affected by socialization processes and individuals' internal dynamics within an ethnic group (*Bernal & Knight, 1993*).

The psychological study of ethnic identity formation has its foundation in the ego identity model of *Erikson (1968)*. According to Erikson, one's identity begins developing in childhood, through a process of "reflection and observation", it becomes particularly prominent during adolescence and early adulthood but may continue through the later stages of adulthood and usually leads to a resolution or an achieved identity (*Erikson, 1968*, p. 22). Individuals who fail a resolution may either withdraw further into social isolation or become lost in the crowd. The empirical study of personal identity was later advanced by *Marcia (1980)*, who theorized identity development as involving two main processes, exploration of identity issues and commitment to significant and related identity domains. Inspired by Erikson's and Marcia's research, developmental psychologist Jean S. Phinney has suggested a three-stage model for ethnic identity development. This model serves as the basis for the present study.

*Phinney (2000)* defines identity as a multidimensional and ever-changing sense of the self in relation to an ethnic group, which is in fact a subgroup within a larger context including culture, race, religion, language, kinship and or place of origin. This sense of the self is not fixed and can change over time and/or different situations (*Herrington et al., 2016*). *Phinney (1992)* holds that with the facilitation of parents and society, youngsters belonging to different ethnic groups have the possibility to develop ethnic identity and that this development is positively related with psychological competence. In sum, *Phinney's (1993)* Developmental Model of Identity applies to all ethnic groups, is based on Marcia's conceptualization of Erikson's theory of identity, and focuses on identity development during adolescence. In subsequent studies, Phinney and others have found that the process of acquiring ethnic identity involves exploration and then affirmation of an individual's identity (*Vedder & Phinney, 2014*). Thus, among young adults, if individuals complete the developmental stages of ethnic identity and affirm their ethnic identity, they grow in self- acceptance *Ivory (2003)*. Importantly, self-acceptance is a necessary factor in one's psychological health and well-being (*Bernard et al., 2013*; *Garcia, Nima & Kjell, 2014*; *MacInnes, 2006*; *Cloninger, 2004*).

In this context, individuals with a highly developed ethnic identity explore their commitments towards their ethnic group (*Ontai-Grzebik & Raffaelli, 2004*); therefore, they recognize their group members, make positive evaluations of their group, have good feelings about being members of the group and perform certain ethnic rituals. On the other hand, individuals with a poorly developed ethnic identity pay little attention to their group members, make negative evaluations of the group to which they belong, do not have a good feeling about being members of the group and do not perform customary commitments to the group (*Phinney, 1991*). Consequentially, ethnic identity and subjective well-being have

been found to have a positive relationship in different ethnic populations (*Herrington et al., 2016*; *Syed et al., 2013*; *Smith & Silva, 2011*; *Krieger, 2010*; *Iwamoto & Liu, 2010*; *Umaña Taylor, 2004*).

## Ethnic Identity and Subjective Well-being

Subjective well-being involves a positive feeling and a general sense of satisfaction with different areas of life (*Myers & Diener, 1995*). According to *Pavot & Diener (2004)*, subjective well-being consists of two distinct, but related components: an emotional component (i.e., positive and negative affect) and a cognitive component (i.e., satisfaction with life). Positive affect reflects the occurrence of positive emotional states such as joy, interest and happiness. In contrast, negative affect involves negative emotional states such as anger, fear, sadness and guilt (*Fredrickson, 2001*). The cognitive component of subjective well-being refers to an assessment of the quality of different aspects of one's life and a general feeling of satisfaction with life (*Gilman, Huebner & Laughlin, 2000*). Life satisfaction, positive affect, and negative affect are considered as independent phenomena (*Lucas, Diener & Suh, 1996*) and are therefore often evaluated individually rather than as an integrated entity (for other perspectives see *Nima et al., 2020a*; *Nima et al., 2020b*; *Garcia et al., in press*).

Several studies have demonstrated a significant positive correlation between ethnic identity and subjective well-being (e.g., *Ali, 2006*; *Diaz & Bui, 2017*; *Oliveira, Pankalla & Cabecinhas, 2012*; *Syed et al., 2013*; *Yoo & Lee, 2005*). There have also been studies indicating that strong ethnic identity can reduce negative emotions (*Williams et al., 2012*; *Yip & Fuligni, 2002*; *Roberts et al., 1999*) and that successful ethnic identity formation can predict an overall sense of well-being (*Vleioras & Bosma, 2005*; *Waterman, 2007*; *Hejazi et al., 2009*). In addition, a significant relationship between identity development and well-being has been demonstrated by few studies conducted in Iran (*Alijani, 2006*; *Ghazanfari, 2003*). Nevertheless, as most studies in this field have been conducted in western cultural contexts, the generalizability of these findings to Iran is limited, since Iran is an Islamic Middle Eastern society with a multiethnic culture. Such culture, has for example, a different approach to gender issues.

## Ethnic Identity and Gender

Research related to ethnic identity and gender has shown mixed results (e.g., *Kazarian & Boyadjian, 2008*; *Ontai-Grzebik & Raffaelli, 2004*; *Phinney, Cantu & Kurtz, 1997*; *Swenson & Prelow, 2005*; *Lum, 2009*). For instance, in a study by Dandy and colleagues (*2008*), no significant gender differences were observed in identity affirmation, but females were more inclined to interact and socialize with other ethnic groups. Other studies, however, indicate higher levels of ethnic identity among males compared to females (e.g., *Vo, 2008*; *Go & Le, 2005*; *Mossakowski, 2018*). Yet other studies show that females score higher than males in ethnic identity (e.g., *Cislo, 2008*; *Creagh-kaiser, 2003*; *Sobansky et al., 2010*).

That being said, at least theoretically, the association between ethnic identity and psychological outcomes, such as life satisfaction and affect, can be mediated by gender. For example, *Cross & Madson (1997)* suggested that since women are more relationship-oriented, their life satisfaction may be positively affected by the sense of belongingness

PeerJ ___________

to their ethnic group. Other studies (e.g., *Navarrete et al., 2010*) have suggested that since men are more commonly the target of racial discrimination, ethnic identity has a larger impact on them compared to its impact on women (*Yap, Settles & Pratt-Hyatt, 2011*). Hence, despite the mixed results regarding gender, besides the relationship between ethnic identity and subjective well-being, we also investigate gender differences in ethnic identity.

## The Multigroup Ethnic Identity Measure (MEIM)

Several models and measures for the assessment of ethnic identity have been proposed. Most of which are based on the works of Erikson, Tajfel, and Phinney (see Umana-Taylor, 2004). Many of the earlier measures were designed for specific ethnic groups, thus, making cross-ethnic comparisons impossible. In order to measure ethnic identity across cultures, *Phinney (1992)* introduced a 20-item ethnic identity measure (i.e., MEIM-O) that included five items to measure Affirmation and Belonging, seven items to measure Ethnic Identity Achievement, two items for Ethnic Behaviors, and six additional items to assess Other Group Orientation (OGO). *Phinney (1992)* considered the two factors, Ethnic Identity and OGO, for MEIM-O based on an exploratory factor analysis done on data collected among American students from different ethnic groups. Later on, in 1999, the original MEIM was revised by Roberts, Phinney, Masse, Chen, Roberts, and Romero, who carried out an exploratory factor analysis on 14 MEIM items (not including the six OGO items). They administered the test to 5,423 young adolescents (sixth to eighth graders) from a variety of ethnicities including African American ($n = 1,237$); Central American ($n = 253$); Chinese American ($n = 177$); European American ($n = 755$); Indian American ($n = 188$); Mexican American ($n = 755$); Pakistani American ($n = 155$); Vietnamese American ($n = 304$); Pacific Islander ($n = 101$); and mixed ancestry ($n = 342$). The results showed that the two items assessing Ethnic Behaviors (i.e., being active in ethnic organizations and participating in cultural practices) were misunderstood among younger adolescents and were therefore removed. Thus, arriving to the version of the MEIM used in the present study; which consists of 12 items and two factors, namely exploration and commitment. More specifically, of the 12 remaining items, five items represent exploration and seven items represent commitment (*Herrington et al., 2016*).[1]

### The MEIM among Diverse Populations

The MEIM in all its multiple versions (including MEIM-O, three-factor 14-item, two-factor 12-item and two-factor 6-item), has been used in dozens of studies and has consistently shown good reliability, typically with alphas above .80 across a wide range of ethnic groups and ages. Table 1 shows the reliabilities of the two-factor 12-item version of the MEIM in several studies across different populations. As shown by Roberts and colleagues' work (*1999*), including a factor analysis of a large sample of adolescents, it seems that the measure can be considered as comprising of two factors: (1) ethnic identity search (a developmental and cognitive component) and (2) affirmation, belonging, and commitment (an affective component; (*Pegg & Plybon, 2005*)). To the best of our knowledge, despite the existence of identity studies among different Iranian ethnicities, no published study has tested the factorial validity of the 12-item version of MEIM in a sample drawn from a Middle Eastern population.

[1] There is also a shorter version of MEIM developed by *Phinney & Ong (2007)*, the Multigroup Ethnic Identity Measure–Revised (MEIM–R) which consists of six items, three of which are related to Exploration and three items to Commitment. These six items were all derived from the 12-item MEIM except for one new item for Exploration: "I have often done things that will help me understand my ethnic background better".

Habibi et al. (2021), *PeerJ*, DOI 10.7717/peerj.10752

**Table 1** Studies validating psychometric properties of different versions of the MEIM in different countries and populations.

| Country | Author | Participants | Ethnicity/Race | EFA/CFA | Reliability (Alpha) |
|---|---|---|---|---|---|
| USA | *Roberts et al. (1999)* | $N = 5,423$ Students in sixth to eighth grades 49% female, Mage $= 12.9$ | African American ($n = 1,237$); Central American ($n = 253$); Chinese American ($n = 177$); European American ($n = 755$); Indian American ($n = 188$); Mexican American ($n = 755$); Pakistani American ($n = 155$); Vietnamese American ($n = 304$); and Pacific Islander ($n = 101$); and mixed ancestry ($n = 342$). | EFA | Total: .85 Factor1: (affirmation, belonging, and commitment)= .84 Factor2 (ethnic identity search): .70 |
| Australia | *Dandy et al. (2008)* | $N = 485$ 55% female Mage $= 12.20$ and SDage $= 1.36$ | Anglo-Celtic Australian ethnocultural background (46%); Asian ($n = 42$); Southern European ($n = 48$); Indigenous Australian ($n = 15$) | CFA (three-factor) | Affirmation:.83 Exploration: .74 OGO: .66 |
| Lebanon | *Kazarian & Boyadjian (2008)* | $N = 525$ 60.4% female Mage $= 16.8$ and SDage $= 1.06$ | Armenian or hyphenated Armenian (e.g., Lebanese-Armenian) | EFA | .88 Ethnic identity $= .70$ affirmation, belonging, and commitment $= .86$ |
| Spain | *Esteban, Nadal & Vila (2010)* | $N = 662$ 55% female and 45% male Mage $= 21.7$ and SDage $= 2.65$ | 326 mestizos and 336 indigenous | EFA and CFA | Overall (For ethnic minority and ethnic majority group): .84 and .83 Ethnic Identity Affirmation:.81 and .79 Ethnic Exploration subscale: .76 and .75 |
| USA | *Burrow-Sanchez (2014)* | $N = 106$ 91.5% male Mage $= 15.5$ and SDage $= 1.3$ | Bicultural with a slight Anglo orientation (44.3%) or balanced bicultural to Mexican oriented (42.9%) | CFA | .99 |
| USA | *Yap et al. (2016)* | $N = 9,107$ 72.8% female Mage $= 20.31$ and SDage $= 3.38$ | Black (8.3%) White (60.9%) Asian (13.7%) Hispanic (14.8%) Middle Eastern (1.3%) Colored–South African (.1%) | CFA | |
| USA | *Sarno & Mohr (2016)* | $N = 622$ LGB self-identified male ($n = 278$), female ($n = 328$), Transgender male-to-female ($n = 3$), Transgender female-to-male ($n = 8$); not reported ($n = 5$). ($M = 22.95$; SD $= 5.64$) | African American/Black ($n = 29$), Asian American/ Pacific Islander ($n = 33$), Latina/Latino/Hispanic ($n = 62$), Native American/American Indian ($n = 21$), or White/Caucasian/European American ($n = 514$). lesbian ($n = 197$), gay ($n = 244$), or bisexual ($n = 181$). | EFA and CFA | Total $= .91$ Clarity $= .75$ Engagement $= .83$ Pride $= .90$ |

## Prior Research on Ethnic Identity in Young Iranian Adults

For instance, *Moghaddas Jafari, Sheikhavandi & Sharif Pour (2008)* studied 347 Kurdish university students of Saqqez Payam-e-Noor University, a Kurdish university located in the west of Iran. They found that Kurdish students were to a large extent committed to all components of ethnic and national identity. They also showed that perceived social inequality was the most important factor explaining and predicting commitment to ethnic and national identity. It is worth mentioning that Iranian ethnic groups reside mostly in the borders of Iran, but also beyond the borders. For example, Kurds partly dwell in Iran and partly in neighboring countries including Turkey and Iraq. Therefore, it has long been a concern of the Iranian central government to maintain the unity of the country in order to strengthen the central power and national security. In this regard, over different eras, political leaders have tried to promote the common aspects of the Iranian culture as opposed to different cultures of ethnic groups within the country. Quite often this has led to the negligence or even suppression of ethnic communities (*Amanolahi, 2005*; *Saleh, 2013*). The national media in Iran, for instance, mostly promulgates the Persian and Shi'i values and culture while the global media emphasizes cultural integration and globalization (*Gulyas, 2017*; *Saleh, 2013*). Indeed, modernization and globalization tendencies have, in certain ways, notably blurred the distinctive characteristics of different ethnic groups. Studies show that, although in recent years there have been several measures taken in Iran to appreciate ethnic differences, for example by teaching Kurdish and Turkish language courses at certain universities, there is still a long way to go before satisfactory conditions are created, where all Iranian ethnicities and sub-cultures are respected and appreciated (*Gulyas, 2017*; *Saleh, 2013*).

Nevertheless, *Hajiyani (2008)* demonstrated that both ethnic and national identity are strong and prominent among the six main Iranian minorities including Turks, Kurds, Lurs, Balochs, Arabs and Turkmen. They also found a significant positive relationship between the sociocultural dimensions of ethnic identity and the cultural and somewhat social dimensions of national identity. Despite the importance of these studies, a major problem is that a majority, if not all, have not used standard models and measures when assessing ethnic identity. This makes comparisons across and within cultures impossible. Standard measures for ethnic identity assessment, such as the MEIM, are under attention in other countries, making research findings comparable.

## The present study

It is important to notice that Iran is a very diverse country with seven main distinctive ethnicities: Farses with 61% of the population are the largest group, Azerbaijanis (16%), Kurds (10%), Lurs (6%), Arabs (2%), Balochs (2%), and Turkmens and Turkic tribes (2%). In addition, there is very small number of Armenians, Assyrians, and Georgians (*World Population Review & Iran population, 2020*). All these ethnicities are referred to as "Iranian", since they are basically from Iran and over the centuries, they have inhabited the Iranian land. At the same time, due to modernization, industrialization, and other economic and political issues, the country has experienced an increase in internal migration patterns,

which have in turn resulted in cities becoming more and more multi-ethnic over the last few decades.

Despite the fact that the MEIM has been extensively used in research around the world, to the best of our knowledge, no psychometric research has been done on the 12-item MEIM in the Iranian young adult population. Additionally, the 12-item MEIM has not been used for assessing ethnic identity in different Iranian ethnic groups. Therefore, this study aimed to confirm the two-factor structure of the Persian version of the 12-item MEIM in a sample of young Iranian adults using confirmatory factor analysis (CFA). In addition, we examined gender differences, test–retest validity, and the reliability of the MEIM. In other words, this research explored the cross-national generalizability of the MEIM's factor structure and also provided an additional test of its construct validity and reliability. Parallel to these aims, we also investigated the association between ethnic identity and subjective well-being (positive affect, negative affect, and life satisfaction). In short, the present paper had three main objectives: (1) To evaluate the factorial validity and test-retest reliability of the MEIM among a sample of young Iranian students, (2) To examine ethnic identity, as measured by the MEIM, in relation to subjective well-being in order to test the concurrent validity of the MEIM (3), and to examine possible gender differences in ethnic identity. Regarding subjective well-being, we expected that (a) ethnic identity will be positively associated with positive affect, (b) negatively associated with negative affect, and (c) and positively related to life satisfaction.

## METHOD

### Participants

Participants were considered eligible if they were within 17 to 40 years of age, had at least completed one semester of university studies, and provided written informed consent for participation in the study. Those who reported psychiatric disorders, organic brain disorder, and/or drug dependence were excluded from the study. Sample size estimation was based on the rule of thumb suggested by *Bentler & Chou (1987)*. A total of 500 undergraduate students volunteered to participate in the normative sample, but 50 (10%) students were excluded because they had responded to less than 90% of the questions. Out of the remaining 450 participants, 24 were withdrawn from the study due to noncompliance. Hence, the final sample included 426 students (91.8% of whom reported being single and 54.7% were males). The percentages of participants from different ethnic groups were as follows: 33.1% Turks, 28.9% Farses, 21.4% Lors, and 16.7% Kurds. The average age was 21.98 years and ranged from 18 to 31 years. The students were 60.3% undergraduates and 39.7% graduates and were from three faculties: Social Sciences (45.1%), Foundational Sciences (33.3%) and Engineering (21.6%). Most of the participants lived in dormitory (70.7%) and the remaining lived with their parents or spouses. All participants were offered the option to take the MEIM on one or two occasions. Those who wanted to take the test twice were part of the test-retest subsample and took the test for the second time after a four-week interval. The test-retest sub-sample ($n = 50$; 25 males) had a mean age of 23 years (age range 18–30 years).

## Measures

### Multigroup Ethnic Identity Measure (MEIM)

The MEIM (*Phinney, 1992*) contains 12 items that can be divided into two subscales: (1) exploration and (2) commitment. The items are rated on a four-point Likert-type scale that ranges from 1 indicating "*strongly disagree*" to 4 indicating "*strongly agree*".

### Positive and Negative Affect Schedule (PANAS)

A self-report assessment of the affective component of subjective well-being in which participants are instructed to rate to what extent they generally have experienced 20 different emotions (10 positive emotions and 10 negative emotions) for the last four weeks, using a 5-point Likert scale (1 = *very slightly*, 5 = *extremely*) (*Watson, Clark & Tellegen, 1988*). The 10-item positive affect scale includes adjectives such as strong, proud, and interested. The 10-item negative affect scale includes adjectives such as afraid, ashamed, and nervous. The PANAS is usually used to operationalize the emotional component of subjective well-being and has excellent internal consistency ranging from .84 to .90 as indicated by Conbrach's alpha coefficients (*Cloninger & Garcia, 2015*; see also *Yoo & Lee, 2005*; *Vera et al., 2008*). The internal reliability in the present study for the positive affect scale was .80 and that for the negative affect scale was .87.

### Satisfaction with Life Scale (SWLS)

This scale is a self-report measure that assesses the cognitive component of subjective well-being and consists of 5 items (e.g., "In most ways my life is close to my ideal") and a 7-point Likert scale (1 = *strongly disagree*, 7 = *strongly agree*) (*Pavot & Diener, 2004*). The estimated internal consistency reliability in this study was .82. The Persian version of this instrument has also been previously used (e.g., *Kjell et al., 2013*) and has been found to have a Cronbach's alpha of .89 (for other studies see (*Yoo & Lee, 2005*; *Edwards & Lopez, 2006*; *Vera et al., 2008*).

## Procedures

With permission from its author (personal communication, 2013), the MEIM was translated into Persian and back-translated to English by a research team (a linguist and two mental health experts) fluent in both Persian and English languages (*Guillemin, Bombardier & Beaton, 1993*; *Villagran & Lucke*, 2005*). A primary Persian version of the MEIM was prepared and its clarity was evaluated by means of a pilot study on 30 students (15 females) who were asked to rate the fluency of items from 0 (Not Understandable) to 5 (Completely Understandable). In this pilot study, participants were asked to report any misunderstanding or lack of clarity related to concepts or wording. The most common rating by participants was completely understandable (response option number five), with this response option endorsed at 95% or higher across all items. Thus, participants' responses indicated no need for item revision. The back-translation was done by other mental health and linguistic professionals. The back-translation followed by experts' judgments revealed that, apart from a few minor adjustments, regarding wording and layout, the Persian version of the MEIM was a precise reflection of its original version. Ethical approval for carrying the study was obtained from The Shahid

Beheshti University's ethics board (95-03-211-31609). To carry out the sampling process, using convenience sampling, research assistants at The Shahid Beheshti University invited students to participate in the study. The ones who accepted the invitation, were handed the questionnaires in in their classrooms in a group survey administration format. Before asking participants to sign the written consent form, they were informed that they could withdraw from participating in the study whenever they wanted to do so, they were given brief description about the aim of the study, and informed about the confidentiality of their participation. The three questionnaires were distributed in different orders, that is, in a counterbalanced design to control for order-effect. The questionnaires included the MEIM, the PANAS, the SWLS, and a demographic information sheet (i.e., gender, age, marital status, and educational status).

## Statistical strategy

A CFA using LISREL, version 8.72 (*Jöreskog & Sörbom, 2005*) was applied to examine the two-factor structure of the MEIM (cf. *Phinney (1992)* and *Phinney (1992)* )). This method offers a variety of statistical tests and indices designed to assess the "goodness-of- fit" of identified models (*Mulaik et al., 1989*). For this purpose, in the present study, the goodness-of-fit was evaluated using the following statistics: the goodness-of-fit index (GFI >.9), the adjusted goodness-of-fit index (AGFI >.90), the non-normal fit index (NNFI >.90), the comparative fit index (CFI >.90), the root mean square residual (RMSR <.08), the normal chi-square ($3 > \chi^2/df < 2$) and the root mean square error of approximation (RMSEA) and its 90% confidence interval (<.05: *Breckler, 1990*; *Mulaik et al., 1989*). In short, the extent to which present data was compatible with different models (and modifications) was examined using LISREL, version 8.72 (Joreskog & Sorbom, 2005). The first model specified a one-factor model ($M_1$) in which all 12 items were forced to load on a single global factor (i.e., *Reese, Vera & Paikoff (1998)*); the second model presented a two-factor orthogonal model ($M_2$); the third examined a two-factor oblique model ($M_3$) as reported in the EFA procedure by *Phinney (1992)* and *Phinney (1992)*. The concurrent validity of the MEIM was investigated by correlations between the MEIM scores and the subjective well-being scores. To evaluate the test-retest reliability of the MEIM, Intra-class correlation coefficients, for the total scale and the two sub-scales, were calculated on two occasions over four weeks apart. Cronbach's alpha and mean inter-item correlation coefficients were calculated for the total MEIM score and its sub-scales. Preliminary analysis of the data showed that normality was violated. The Z score for the univariate skewness values ranged from −4.40 [Item 9, "I have a lot of pride in my ethnic group"] to −.76 [Item 1, "I have spent time trying to find out more about my ethnic group, such as its history, traditions, and customs."] (See Table 2), and Relative Multivariate Kurtosis was 1.16 and test of multivariate normality for skewness and kurtosis was $\chi^2 = 255.29$, $p < .001$. Due to univariate and multivariate non-normality of the MEIM items, the weighted least squares (WLS) estimation method was applied in CFA (*Bentler & Bonett, 1980*).

Habibi et al. (2021), *PeerJ*, DOI 10.7717/peerj.10752

**Table 2  Parameter estimates and goodness-of-fit indexes for CFA of the MEIM.**

| Items | Z | P.E$_T$ | P.E$_m$ (P.E$_f$) |
|---|---|---|---|
| commitment | | | |
| 3- I have a clear sense of my ethnic background and what it means for me. | −1.29 | .70** | .70**(.70**) |
| 5- I am happy that I am a member of the group I belong to. | −4.04 | .81** | .86**(.73**) |
| 6- I have a strong sense of belonging to my own ethnic group. | −3.26 | .89** | .76**(.80**) |
| 7- I understand pretty well what my ethnic group membership means to me. | −1.48 | .82** | .86**(.77**) |
| 9- I have a lot of pride in my ethnic group. | −4.39 | .85** | .90**(.81**) |
| 11- I feel a strong attachment towards my own ethnic group. | −2.64 | .89** | .95**(.83**) |
| 12- I feel good about my cultural or ethnic background. | −3.17 | .77** | .89**(.67**) |
| 1- I have spent time trying to find out more about my ethnic group, such as its history… | −0.76 | .63** | .64**(.64**) |
| 2- I am active in organizations or social groups that include mostly members of my own ethnic group. | 1.16 | .55** | .62**(.50**) |
| 4- I think a lot about how my life will be affected by my ethnic group membership. | −0.13 | .58** | .64**(.50**) |
| 8- In order to learn more about my ethnic background, I have often talked to other people about my ethnic group. | −1.11 | .79** | .76**(.84**) |
| 10- I participate in cultural practices of my own group, such as special food, music, or customs. | −2.42 | .75** | .74**(.78**) |

| Model | NNFI | RMSEA | CFI | χ2 (df) | GFI | ECVI | Δχ$^2$ |
|---|---|---|---|---|---|---|---|
| M$_1$ | .65 | .087(.076–.099) | .71 | 229.69(54) | .84 | .65 | – |
| M$_2$ | .28 | .124(.11–.135) | .42 | 407.06(54) | .72 | 1.07 | 229.54** |
| M$_3$ | .69 | .082(.071–.094) | .75 | 205.99(53) | .85 | .60 | 201.07** |
| M$_{3a}$ | .88 | .053(.039–.067) | .90 | 107.62(49) | .92 | .39 | 98.37** |
| M$_b$ | .91 | .051(.34–.066) | .93 | 151.98(98) | .97 | .63 | – |
| M$_c$ | .90 | .054(.039–.069) | .92 | 177.17(109) | .96 | .64 | 25.19** |
| M$_m$ | .87 | .056(.035–.076) | .91 | 84.51(49) | .94 | .61 | 67.47** |
| M$_f$ | .94 | .044(.0095–.069) | .93 | 67.47(49) | .96 | .65 | 84.51** |
| M$_{vc}$ | .90 | .054(.039–.068) | .91 | 181.48(112) | .96 | .64 | - |
| M$_{var\_co}$ | .89 | .055(.040–.069) | .91 | 180.50(110) | .96 | .64 | .98 |
| M$_{var\_ex}$ | .89 | .055(.040–.069) | .91 | 181.39(111) | .96 | .64 | .09 |
| M$_{co\_var}$ | .90 | .054(.040–.069) | .91 | 180.73(111) | .96 | .64 | .75 |

**Notes.**

All parameter estimates were significant at $p < 0.01 =$**, $Z = Z$ score for tests of univariate normality.

P.E$_T$, Parameter estimation for the two-factor oblique model fo total group, and correlated errors; P.E$_m$, Parameter estimation for the two-factor oblique model for male, and correlated errors; P.E$_f$, Parameter estimation for the two-factor oblique model and correlated errors for female; M$_1$, one-factor general model; M$_2$, two-factor orthogonal model; M$_3$, two-factor oblique model; M$_{3a}$, two-factor oblique model and correlated errors; M$_b$, baseline model for boys with diagonal error covariance; M$_c$, model with pattern of factor loadings held invariant; M$_m$, baseline model for boys with diagonal error covariance; M$_f$, baseline model for girls with diagonal error covariance; M$_{vc}$, model with all factor variances and covariances held invariant; M$_{var\_co}$, model with variance of commitment held invariant; M$_{var\_ex}$, model with variance of exploration held invariant; M$_{co\_var}$, model with covariance of commitment with exploration held invariant; CI, confidence interval; ECVI, expected cross-validation index; NNFI, non-normed fit index; RMSEA, root mean square error of approximation; CFI, confirmatory fit index; GFI, goodness of fit index.

## RESULTS

### Aim 1: Factorial Validity and Test-Retest Reliability of the MEIM

Table 2 presents the fit estimates for all models. The one-factor model and the two-factor orthogonal model did not meet the previously specified fit criteria and the two factors oblique model showed inadequate fit to the data ($M_1$ to $M_3$). The modification by correcting errors in the two-factor model revealed some improvement ($M_{3a}$; $RMSEA =$ .053). The chi-square test results were significant for all models, but that is to be expected with models with large degrees of freedom and relatively large sample sizes (*Hu & Bentler, 1995*). An examination of the remaining fit indices (*Satorra & Bentler, 2001*) for nested models suggested that the oblique and correlated errors model ($M_{3a}$ in Table 2) was significantly more acceptable than the two factors oblique model ($M_3$; $\chi^2 = 102.07$; $df = 4$, $P < .001$). In a comparison of the nested models, the $\chi^2$ (*Jöreskog & Sörbom, 1993*) showed that the two-factor oblique correlated errors model provided a better fit [S-B $\chi^2/df =$ 2.19; $CFI = .90$; $NNFI = .88$; and $RMSEA = .053$ ([CI] 90% = .039, .067]. The correlation found between exploration and commitment latent variables was .88 ($p < .001$).

Another series of CFAs were conducted in order to determine if the factorial structure of the MEIM was consistent across gender. In short, testing the factor structure of MEIM in both males and females is the main starting point in evaluating multivariate CFA across genders and building baseline models for each group separately without any specification of equality constrained across groups. For testing equivalency across genders, both baseline models were modified to include the specification of four error covariances for males, and for females. First, a CFA was conducted for males and females separately. Several steps were taken in order to obtain satisfactory model fit and these steps were guided by parsimony and meaningfulness perspectives (Byrne, Shavelson, & Muthén 1989). The comparison of the baseline model ($\chi^2/df = 1.55$; $CFI = .93$; $NNFI = .91$; and $RMSEA = .051$ ([CI] 90% = .034, .066) and the modified models for males ($M_m$; $\chi^2/df = 1.72$; $CFI = .91$; $NNFI = .88$; and $RMSEA = .056$ (90% CI [.035–.076]) and females ($M_f$; $\chi^2/df = 1.38$; $CFI = .96$; $NNFI = .94$; and $RMSEA = .044$ ([CI] 90% = .0095, .069), indicated that the modified models for males ($\Delta^2 = 67.47$, $df = 49$, $p < .05$), and for females ($\Delta^2 = 84.51.4$, $df = 49$, $p < .01$) were significantly different from the diagonal error covariance baseline model. Hence, these models provided an adequate level of fit for males and females as baseline models. Table 2 (model $M_b$) shows that the hypothesized factor structure of MEIM revealed good fit across genders, but the pattern of factor loadings (model $M_c$) varied across genders (Table 2). All constrained models in Table 2 were compared with $\chi^2$ statistics against a starting or baseline model ($M_b$) in which no constraints in estimation of parameters were specified. The results indicated that the variances and covariance of commitment and exploration were consistent across gender.

In the case of configural invariance, $\chi^2$, RMSEA, CFI, NNFI, and other fit indices were used to examine whether or not the combined models had a good model fit. Furthermore, for metric, scalar, residual, and latent variances and covariance, the RMSEA values and RMSEA confidence intervals of the hierarchical (nested) models were compared. For example, in the case of comparison of the weak and strong factorial invariance models, if

**Table 3** Internal consistency coefficients, mean inter-item correlation, means, and standard deviations for males, and females.

| Multigroup ethnic identity | α | Correlation | M | SD |
|---|---|---|---|---|
| Commitment | .91(.88) | .59(.51) | 22.12(21.02) | 4.61(4.34) |
| Exploration | .78(.75) | .41(.38) | 13.68(12.75) | 3.15(3.27) |
| MEIM | .92(.90) | .49(.42) | 35.76(33.81) | 7.29(7.03) |

Notes.
$p < .01 = **$.
MEIM, Multigroup Ethnic Identity Measure, Values for females are inside the parenthesis.

**Table 4** Mean and standard deviation of time 1 and 2 and test-retest reliability of MEIM and subscales.

| Measures | Time1 | | Time 2 | | r |
|---|---|---|---|---|---|
| | M | SD | M | SD | |
| commitment | 23.15 | 5.12 | 23.9 | 4.89 | .91 |
| exploration | 13.89 | 3.2 | 14.45 | 3.4 | .82 |
| MEIM (Total Score) | 35.89 | 6.15 | 36.15 | 7.01 | .86 |

Notes.
Note- intra-class correlation coefficient, all $p$ values $< .01$.

RMSEA values fall within one another's confidence intervals, this shows strong factorial invariance. Then, the changes in the CFI of hierarchical (nested) models were examined. Also, the change in CFI for the weak and strong factorial invariance models was assessed. A change of less than .01 shows strong factorial invariance (*Cheung & Rensvold, 2002*). That is, it shows equal form (i.e., the number of factors and the pattern of factor-indicatorrelationships are the same), equal factor loadings, equal thresholds (i.e., when observed scores are regressed on each factor, the thresholds are equal), and equal residual variances across gender.

The internal reliability coefficients and the mean inter-item correlation for the MEIM are presented for the total sample and separately for males and females in Table 3. These findings suggest that the scale has acceptable internal consistency. For test-retest reliability of the MEIM, 50 Students (15 males) completed the scale with four weeks between measurements (i.e., time 1 and time 2). Intra-class correlation coefficients between the total and subscale scores at time 1 and 2, respectively, ranged from .82 to .91 (see Table 4).

## Aim 2: Concurrent Validity of the MEIM

Table 5 presents the Pearson correlation coefficients between MEIM and its subscales with positive affect, negative affect, and life satisfaction. The results indicate that, as expected, the MEIM and its subscales have a positive relationship with positive affect and Life satisfaction and a negative relationship with negative affect. The shared variance for the total score of the MEIM indicates weak relationships between ethnic identity and subjective well-being measures.

**Table 5** Correlations between the ethnic identity and its subscales and subjective well-being including positive affect, negative affect, and life satisfaction ($n = 426$).

| Measures | Positive affect | Negative affect | Life satisfaction |
| --- | --- | --- | --- |
| Ethnic identity | .28[**] | −.29[**] | .31[**] |
| commitment | .28[**] | −.25[**] | .32[**] |
| exploration | .23[**] | −.27[**] | .25[**] |

**Notes.**
[**]$P < .01$ (two-tailed).

## Aim 3: Gender Differences and Ethnic Identity

A noticeably different pattern of results was found for males and females (Table 3). Female students had a significantly higher score on the MEIM compared to male students ($t(424) = 2.71$, $p = .007$). A multivariate analysis of variance (MANOVA) was conducted to evaluate the effects of gender on the MEIM sub-scales, with Bonferroni adjustment for multiple comparisons (*Huberty & Morris, 1989*). The MANOVA results showed a significant effect: Hotelling's Trace $= .02$, $F(3, 1158) = 4.33$, $p = .014$, $\eta^2 = .02$ (Table 3). Subsequent examination of between-subject effects showed that the female group scored significantly higher on exploration ($F(1, 402) = 7.50$, $p = .003$, $\eta^2 = .018$) and commitment ($F(1, 402) = 4.99$, $p = .016$, $\eta^2 = .014$).

## DISCUSSION

The purpose of this study was to confirm the factor structure, reliability, test-retest and concurrent validity of the Persian version of the MEIM in a sample of young Iranian adults, and to see if there are any gender differences of ethnic identity in the sample. Generally, the scale was found to be reliable and valid for young adults and both genders. This study serves an important purpose since ethnic identity plays a major role in normal development and positive youth adjustment (*Rivas-Drake et al., 2014*). Although a lot of research has been done on ethnic identity, the majority of these studies have focused on different ethnic groups within the American population and context (e.g., *Rivas-Drake et al., 2014*; *Roberts et al., 1999*; *Yap, Settles & Pratt-Hyatt, 2011*). Due to the fact that a variety of ethnicities exist in Iran, and also because a significant population in the world speak Persian including Iranians, Afghanis, Tajikistanis, and also many immigrants in other countries such as USA, UAE, Canada and Turkey, we found it important to investigate the psychometric properties of the Persian version of the 12-item MEIM. The results indicated that a two-factor oblique with correlated errors model ($M_{3a}$) fit the data better than the two other models ($M_1$ to $M_2$). All 12 items were loaded on their respective factors, and factor loadings ranged from .54 to .78 for the exploration factor, and from .71 to .90 for the commitment factor. This suggests that the best way to interpret the MEIM scores would be to consider one global score and two sub-scores (i.e., exploration and commitment). Indeed, a similar hierarchical model was suggested by *Phinney & Ong (2007)*, as a model with generally acceptable (not the best) fit criteria for the revised version of the MEIM in English (MEIM-R).

The Cronbach's alpha coefficients (.75–.92), mean inter-item correlations (.38–.59), and intra-class coefficients between two points in time with a four-week interval (.82–.91), all

support the reliability of the Persian version of the MEIM across genders. These findings are consistent with previous research indicating good internal reliability for the MEIM in other cultures (*Cuellar et al., 1997*; *Dandy et al., 2008*; *Lee et al., 2001*; *Phinney, 1992*; *Ponterotto et al., 2003*; *Spencer et al., 2000*; *Yip & Fuligni, 2002*).

Moreover, the construct validity of the MEIM was supported by the relatively modest, albeit statistically significant, correlations between both its total and subscales scores and measures of subjective well-being (e.g., positive affect, negative affect and life satisfaction). All correlations were in the expected directions, thus, replicating earlier studies (e.g., *Beiser & Hou, 2006*; *Dandy et al., 2008*; *Juang & Syed, 2010*; *Kiang et al., 2006*; *Martinez & Dukes, 1997*; *Phinney & Ong, 2007*; *Roberts et al., 1999*). More specifically, consistent with previous findings, we found positive relationships between ethnic identity (both total and subscales scores) and both positive affect (cf. *Dimitrova et al., 2013*; *Juang & Syed, 2010*) and life satisfaction (cf. *Dimitrova et al., 2013*; *Dimitrova, Ferrer-Wreder & Trost, 2015*; *Dimitrova, Johnson & Van de Vijver, 2018*; *Juang & Syed, 2010*; *Williams et al., 2012*), while ethnic identity (both total and subscales scores) was found to be negatively related to negative affect (*Abu-Rayya, 2006*; *Beiser & Hou, 2006*; *Dimitrova et al., 2013*; *Juang & Syed, 2010*). Given that ethnic identity is only one of the many correlates of subjective well-being, the relatively small effect size in these correlations seems rational. However, regarding the construct validity issue, the directions of the relations are of particular significance.

With regard to gender, we found significantly higher scores in ethnic identity and its two components (exploration and commitment) in women compared to men. This finding is in line with some of the previous research (e.g., *Dion & Dion, 2001*; *Juang & Syed, 2010*; *Suárez-Orozco & Qin, 2006*). Gender differences may lie in the fact that women more commonly feel that they are expected and even pressurized to preserve their cultural and ethnic values (*Juang & Syed, 2010*; *Yip & Fuligni, 2002*). Indeed, in the Iranian culture women are often told to act like a "*Khanoum*", which means "lady", which is an expression mirroring society's expectation that women should be committed to and act in congruence with sociocultural norms and pre-defined values (*Rashidian, Hussain & Minichiello, 2013*). Such expectations can influence Iranian females define themselves by their ethnic identity and too seek the validation of their ethnic groups and the broader society in order to be honored with the "noble" title of "*Khanoum*". Furthermore, in the Iranian context, when a female deviates from their ethno-cultural values and codes of behavior, the repercussions they face are far greater than that a male facing the similar situation—for a female, this might mean losing her family support altogether. The need to receive the family's validation and acceptance is, for instance, one of the important needs expressed by Iranian adolescent girls, as suggested in the qualitative study conducted by Mousavi and her colleagues among 27 Iranian adolescent girls (*Mousavi et al., 2018*).

## CONCLUSIONS, LIMITATIONS, AND FUTURE DIRECTIONS

In conclusion, the results from this study provide support for the validity, reliability and factor structure of the Persian version of the 12-item MEIM in a young adult sample. The results generally support the use of the 12-item MEIM among Iranian young adults. We propose that the MEIM is specially suitable for research on ethnic identity in Iran which is a multiethnic country; since the MEIM is a widely used instrument and it was designed to be utilized across different ethnic groups (*Phinney, 1992*). Hence, its validation in an Iranian context was of great importance. It is also very useful for comparing variances in ethnic identity and well-being among people from different countries that, for one reason or another, live in another country as immigrants. Moreover, the validation of the MEIM in the Iranian context facilitates the comparison of research findings to those from other cultures in which this widely spread instrument has been used.

However, since ethnic identity gradually develops during adolescence and young adulthood through the processes of exploration and commitment (*Roberts et al., 1999*), further empirical work needs to be conducted to validate the factorial analyses in a sample of younger adolescents (e.g., 12-17 years old). In this vein, since the population in this study was limited to older adolescents (i.e., 17 years old and above), we recommend that researchers should use the 12-item Persian version of the MEIM with caution on younger populations. This is of special importance if the research issues are related in any way to gender; since our study shows that gender may play an important role as a mediator variable. It is worth noting that the 6 items MEIM-R version of this instrument is also confirmed by *Phinney & Ong (2007)* as a tool that covers the core aspects of ethnic identity. Therefore, further studies can look into analyzing the psychometric properties of the 6-item MEIM-R in the Iranian context, in order to have a version for faster administration. In addition, there is a need to investigate the factor structure of ethnic identity within ethnic minority groups in Iran, which was not possible in this study due to the small numbers of participants in each ethnic group. Future studies may also include investigations of the OGO-scale and of the EI-factor structure.

That all being said, the findings of this study provide useful information, adding to the existing literature on ethnic identity in Iran. Hence, laying the groundwork for further studies on other aspects of ethnic identity in the Iranian context and other ethnic groups around the world.

### Funding

The development of this article was funded by a grant from the Swedish Research Council (Dnr. 2015-01229). The funders had no role in study design, data collection and analysis, decision to publish, or preparation of the manuscript.

### Grant Disclosures

The following grant information was disclosed by the authors:
The Swedish Research Council: Dnr. 2015-01229.

## Competing Interests

The authors declare there are no competing interests.

## Author Contributions

- Mojtaba Habibi conceived and designed the experiments, performed the experiments, analyzed the data, prepared figures and/or tables, and approved the final draft.
- Maede Sadat Etesami performed the experiments, authored or reviewed drafts of the paper, and approved the final draft.
- Mohammad Ali Taghizadeh performed the experiments, analyzed the data, prepared figures and/or tables, and approved the final draft.
- Faezeh Sadat Akrami performed the experiments, prepared figures and/or tables, and approved the final draft.
- Danilo Garcia conceived and designed the experiments, authored or reviewed drafts of the paper, and approved the final draft.

## Human Ethics

The following information was supplied relating to ethical approvals (i.e., approving body and any reference numbers):

The University of Shahid Beheshti's ethics board approved this study (95-03-211-31609).

## Data Availability

Raw data are available in the Supplemental Files.

## Supplemental Information

Supplemental information for this article can be found online at http://dx.doi.org/10.7717/peerj.10752#supplemental-information.

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
