# Peer review of "Psychometric properties of the Multi-group Ethnic Identity Measure (MEIM) in a sample of Iranian young adults"

_PeerJ, doi:10.7717/peerj.10752_

## Round 0.1 · original submission · Major Revisions

I have read your article and also been fortunate to receive three detailed reviews from three experts in your field (note that Reviewer 3 provided additional feedback by annotating your article directly - please find that attached). From their reviews, as well as my own reading of you article, I do not believe that your manuscript is currently suitable for publication but that with revision it likely will be.

All three reviewers provided thoughtful feedback and I encourage you to respond carefully to each of their points as you prepare your article for re-submission. I will not reiterate each of the reviewers' points here but I did want to highlight some common elements from their reviews:

All three reviewers commented that your Introduction would benefit from you tightening the focus of it and clarifying your aims. Additionally, reviewers 1 and 3 suggested that you should more-carefully describe previously-used scales in your introduction and how your methods relate to them. Additionally, reviewers 1 and 3 also called for a more thoughtful consideration of gender in relation to your findings. Finally, and perhaps most importantly, both reviewers 1 and 3 raised some concerns with your methods and analyses. Many of these can likely be addressed by enhancing the clarity of your reporting, but others may require you to reconsider your analytical approach.

Thank you for considering PeerJ. I look forward to receiving your revision.

·

Basic reporting

This manuscript consists of a report on the factor structure and correlates of the MEIM in an Iranian sample. As the MEIM is a widely used measure of ethnic identity, it is certainly valuable to assess its psychometric properties in a wide variety of samples. In this sense, the manuscript makes a useful contribution.

There are, however, several issues that weaken that contribution, as elaborated below.

The Introduction could be tightened up considerably and focused on the issue at hand. For example, the discussion of the three-stage model of ethnic identity is not only rather outdated, but it has little bearing on the current investigation. The rationale for this study is rather straight-forward and can be communicated quickly to readers.

There were two major problems with the Introduction. First, related to my previous point the Introduction does not rely on the latest research and thinking. For example, the Ethnic/Racial Identity Study Group published four useful papers in Child Development in 2014 covering conceptual and methodological issues, as well as a meta-analysis (COI: I was part of that group and an author on those papers). Additionally, there is quite a bit of psychometric work with the MEIM, including two papers by Yap et al. (2014, 2016) and a reliability generalization paper by Herrington et al. (2016). All of this aforementioned literature is more central to the current study than what was actually included.

The second major issue is that there was insufficient attention paid to the fact that there are multiple versions of the MEIM. In the authors’ review of past work, they do not attend to which studies relied on each version. This is important because the original version described in Phinney (1992) was 14 items and tends to have a three-factor structure, whereas the Roberts at al. (1999) revision, which is the one used in this paper, has 12 items and tends to have two factors. There is also the 6-item Phinney & Ong (2007) version. At the beginning of the Discussion section the authors mention inconsistencies in the past literature, but they do not recognize that they are comparing different scales, not to mention that most others did not compare multiple models and/or they only relied on EFAs not CFAs. In sum, there is quite a lot of psychometric work on the MEIM in the literature, yet the current study scantly references that work nor does it use it to guide the current investigation.

The focus on gender was unclear and under-theorized. By and large past research has not found many gender differences and there is not a compelling reason for why we would observe them.

With regard to relations to well-being there are meta-analyses on the topic that should be discussed (Smith & Silva, 2011; Rivas-Drake et al., 2014).

The raw data that were submitted only included the 12 items of the MEIM and not any data for the correlates.

In Table 1, some data appear to be missing (the upper bound of the RMSEA interval; the df for the models). Moreover, it seems like something is not right if the orthogonal model and oblique model have the same CFI, given the substantial correlation between the two.

What errors were correlated and why? Just because of modification indices? What meaning does this have?

Why is N only 205 for the correlations with well-being? Are those manifest scale scores and not latent variables? Why not use the latent variables identified through the factor analysis?

Experimental design

The authors reported that the sample included four ethnic groups. It seems like conducting invariance testing across the four groups would be useful vis-à-vis the goals of the study.

The test-retest sample seems quite small, and in the text focuses on total score (although the table also included the subscales). I wonder about the generalizability of this finding given the nature of the sample (That said, my knowledge of how this scale operates is consistent with this finding).

The authors use MANOVAs to test for gender differences, apparently only as a gatekeeper F to testing the univariate effects. As described by Huberty & Morris (1989) this is not a proper use of MANOVA. If Type 1 error is a concern then the proper action would be an alpha correction.

Validity of the findings

Overall I think the data are useful and important to get out there, but that the manuscript needs quite a bit of work. I am also left wondering what the authors recommend in terms of using this scale in an Iranian sample. Must it be used in latent space with the hierarchical model? This is where my earlier question about the correlates comes in; those were (best I could tell) correlations based on manifest variables. What would they look like using the latent variables?

Additional comments

No additional comments

Reviewer 2 ·

Basic reporting

This article documents a substantial forward advance in the study of identity formation, in particular, ethnic identity formation. The adaptation of Phinney's construct and measures for translation into Persian, along with the validation of the measure's essential factor structure, bring forward a new tool that will be useful for the study of identity development in groups which have previously been excluded. The need for the study was persuasively articulated.

The number of references for the article (and the number of in-text citations) is a bit excessive, and this makes the writing somewhat cumbersome at times. I suggest that the authors review their in-text citations and try to eliminate two or more pages of references (particularly old references, unless they are to seminal studies). Where a recent paper builds upon previous studies, it is the only one that needs to be cited and referenced.

Experimental design

The research question and problem were well defined. The sample size was adequate for testing the factor structure of the measure. Appropriate steps were taken for the translation of the measure. The design and report of the factor analysis were clear.

Validity of the findings

Implications of the study were outlined and appropriate conclusions were drawn.

Additional comments

This study represents important work. I encourage the authors to continue to build upon this advance in order to study identity development in heretofore unstudied populations.

·

Basic reporting

I felt this study would have benefited from better organization of the introduction to reflect the objectives addressed by the paper, and to focus in on the key constructs addressed in the paper. I appreciate that there is a wide range of literature covered and the authors were comprehensive in their writing, but narrowing it down would help make the focus of the paper more clear.

In the methods section, there are some inconsistencies in the reporting on the number of participants. There doesn't appear to be a mention of the total number of participants approached for the study in order to arrive at the final sample - no response rate is provided. It is also unclear if the twenty-four forms excluded from the study were done so by participant request or reviewer decision (two different explanations are provided in the methods section). The section on measures also included some information that would be better suited to the introduction.

The reporting in the results section was clear and comprehensive.

There are some areas where the writing could be strengthened, usually with small grammatical errors such as use of plural when the term is singular. This is primarily seen in the discussion section.

I have made notes in the manuscript with specific comments or suggestions when warranted.

Experimental design

The research question is well-defined and the investigation as described was rigorous. The only aspect I am unable to judge at the moment is the quality of the process for recruitment of participants, as more information is needed on how recruitment was carried out, the rate of participation, and the reason for the excluded responses. See my comments in lines 264-271.

Validity of the findings

Though the conclusion as written is linked to the original question, I would have liked to see additional discussion of the interpretation of the results (as the current focus now is primarily on a summary/overview of results). One area for discussion may be how these results are related to the Iranian context. The authors make some comments (e.g., that gender differences may relate to cultural expectations; that the MEIM is particularly well-suited to multiethnic populations such as that found in Iran) that would benefit from some additional discussion, as they appear very relevant to the stated goals of the article and would be a useful contribution.

Additional comments

This is an overall strong study that would benefit from streamlining the introduction, addition of relevant material on recruitment in the methods section, and further contextualizing the results in the discussion.

---

## Round 0.2 · Major Revisions

Thank you very much for submitting your revision to PeerJ. I apologize that it took me a while to identify reviewers to review your article and unfortunately I was not able to get any of the three original reviewers to review this revision. Therefore, two new reviewers have reviewed your revised submission, but both were aware that this was a revised submission. In spite of this, both reviewers have some additional feedback and requests for clarification that they pose to you, especially with regards to your framing and your analytical approach. However, I believe that these reviews are much less major than your previous round and so I look forward to receiving your revised article.

Reviewer 4 ·

Basic reporting

The manuscript is somewhat difficult to read, possibly because English is not the first language for the authors; at a minimum, the authors should run the entire paper through spell-checking (e.g., Phinney's name is misspelled in one instance) with grammar checking enabled. In terms of literature references, given that Phinney's entire program of research is based upon Erik Erikson's ego psychology, the authors need to read Erikson directly and include one or more citations of his theory at the beginning of the paper (Erikson writes about identity in general, as well as ethnic identity in particular); near the end of the paper, it would be useful for the authors to acknowledge that Phinney and colleagues currently recommend the 6-item Revised MEIM (Phinney & Ong, 2007), rather than the older version that the authors have used.

Experimental design

The research question is fairly well-defined (i.e., Can the two-factor pattern of the 12-item MEIM be replicated in a sample of Iranian men and women), although the analyses do not quite match the question (see additional comments under Validity of the Findings, below).

Validity of the findings

Based upon the Introduction, I would have expected the authors to run two versions of a multiple-group confirmatory factor analysis (with gender as the group): (1) Two-factor solution, constrained to be equal across genders; followed by (2) two-factor solution, allowed to vary across groups -- not the series of analyses that apparently were intended as a step-by-step comparison of various models that Roberts et al. (1999) had conducted. Also, the rationale for using weighted least squares (WLS) estimation is questionable; in the absence of stronger evidence for non-normality at the univariate level (e.g., skewness as well as kurtosis with an absolute value of 2.30 or higher; see Lei & Lomax, 2005) and at the multivariate level (e.g., relative multivariate kurtosis of 1.96 or higher), I would suggest using the default (i.e., maximum likelihood) extraction.

Additional comments

I believe that this paper deserves further consideration. However, I do not believe that the current version should be published. Although I am aware that the authors have addressed previous reviewers' comments, I thought that I should let the authors know about examples of criticism that they may encounter, in the event that the paper is published. Overall, I would be interested in reviewing a substantially revised version of this manuscript.

Reviewer 5 ·

Basic reporting

In my view major parts are to be entirely revised as argumented below.

- The introduction focuses on adolescence rather than emerging/established adults.
- The whole text needs careful revision for language and style (too long paragraphs etc.) and be more concise on main topics (identity, MEIM, well-being and gender by convincing what is novel, why the need for this study and describe well the local context). What do we know about these topics in Iran, what studies and relevance? Why the need of this work in Iran?
- More recent references as many are too old (2008, 1992 etc.)
- Ethnic identity different definitions do not review these and one may wonder why these are needed?

Experimental design

- I have a major issue with the large age range that needs a justification and split (emerging adults or established adults).
- I similar major concern for the methods and analytic steps. I would strongly advise to introduce measurement invariance for gender and longitudinal testing as mean differences cannot be compared in absence of scalar invariance. Then, a model linking identity and well-being in a path model using SEM is needed.

Validity of the findings

I would advise to run more sophisticated analyses and make a stronger contribution both conceptually and methodologically.

Additional comments

- Why Table 1 reports other MEIM studies? Is this a review paper?
- Why Table 2 reports indices for single item and not for MEIM factors and gender?
- The figure needs to be drawn from scratch rather than a copy from an output and added one for gender.

---

## Round 0.3 · accepted · Accept

Thank you for resubmitting your article to PeerJ. I believe that you have thoroughly responded to the latest round of reviewer feedback and I did not feel the need to send your revised article back out for review. Therefore, it is my pleasure to recommend your article for publication.